# Permeability and Leaching Properties of Recycled Concrete Aggregate as an Emerging Material in Civil Engineering

**Andrzej Głuchowski [1],\*** , **Wojciech Sas [1], Justyna Dzięcioł [1], Emil Soból [2] and Alojzy Szymański [2]**

[1] Water Centre Laboratory, Faculty of Civil and Environmental Engineering, Warsaw University of Life Sciences, 02-787 Warsaw, Poland; wojciech_sas@sggw.pl (W.S.); justyna_dzieciol@sggw.pl (J.D.)

[2] Department of Geotechnical Engineering, Faculty of Civil and Environmental Engineering, Warsaw University of Life Sciences, 02-787 Warsaw, Poland; emil_sobol@sggw.pl (E.S.); alojzy_szymanski@sggw.pl (A.S.)

\* Correspondence: andrzej_gluchowski@sggw.pl; Tel.: +48225-935-405

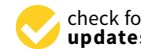

**Featured Application: Recycled concrete aggregate (RCA) is an emerging alternative material as a substitute for natural aggregate. The RCA blends test results in this paper has shown the appropriate coefficient of permeability for earth dam construction and for sub-base layers in road engineering. Moreover, the RCA is environmentally safe.**

**Abstract:** In this article, a study of the threshold gradient and leaching properties for recycled material, namely, recycled concrete aggregate (RCA), was conducted. The RCA in this study is a material that comes from recycling concrete debris. A series of tests in permeameter apparatus in a constant head manner were conducted. The test method has been improved to eliminate common mistakes, which occur when the constant head method is used. During the following study, aggregates with gradations equal to 0–8, 0–16, and 0.05–16 mm were tested. The tests were conducted on gradients ranging from 0.2 to 0.83. This range of tested gradients led to the evaluation of the flux velocity and indicated non-Darcian flow. For engineering applications, the threshold gradients for three RCA blends were calculated using a statistical analysis. The average coefficient of permeability, $k_{avg}$, for linear flow was equal to $1.02 \times 10^{-4}$–$1.89 \times 10^{-4}$ m/s. In this paper, suffosion analysis was also conducted for the three blends in order to eliminate the possibility of particle movement. Moreover, for RCA blend 0–16 mm, leaching properties was examined. It was found that the concentration of chlorides, sulphates, and heavy metals in the water solution does not exceed the permissible standards. This paper ends with conclusions and proposals concerning the threshold gradients obtained from the statistical analysis, suffosion analysis, and flux velocity.

**Keywords:** permeability; recycled concrete aggregate; threshold gradient; constant head method; coefficient of permeability; leaching properties

---

## 1. Introduction

Increasing economic growth in Central Europe results in a growing waste problem. The development of recycling procedures, composting, and incineration facilities seem to prove the problem's existence. However, construction wastes constitute a large percentage (more than 50%) of solid waste stored in landfills [1]. Moreover, the failure to recycle such materials results in environmental disruption through disposing landfill, which is mostly unreasonable [2,3]. The industrial sector, associated with the construction of buildings and the demolition of existing structures, generates about 868.5 million

tons of waste, which amounts to 34.7% of total waste production in the entire European Union [4]. At that time, 17.0 million tonnes of this waste were produced in Poland, which is 9.5% of the total [4]. To solve growing problem of construction and demolition waste, methods that would make it possible to reclaim construction materials are strongly desired. The life cycle of concrete, which ends with the demolition phase, is significant in the recycling process and should be handled with care because of potential reuse of the crushed concrete [3]. Therefore, knowledge about every possible property of demolished concrete is necessary and strongly demanded.

## 2. Literature Review

Aggregates are materials commonly used in civil engineering. For example, in the European Union and the EFTA countries in 2016–2017, around 30,000 tons of aggregates are used to create 1 km of new road [5]. These are mainly natural aggregates—such as crushed rocks, gravel, and sand—which constitute about 88% of the market demand [5]. Moreover, natural aggregates are the only material used in construction like earth dams, embankments, levees, or any others earthen construction. Designers and contractors are wary of using recycled aggregates, because the physical and mechanical properties are different from the natural aggregate behavior. The lack of detailed knowledge about recycled aggregates properties is another issue [6]. Despite this, in the last four years, the demand for recyclable aggregates increased from 5% to 8%, which is a significant increase on the scale of the European Union and EFTA (The European Free Trade Association) countries [5,6]. The largest number of recycled aggregates is produced in the UK, 52.3 million tonnes and in France 20.3 million tonnes compared to Poland, 5 million tonnes according to data for 2015, published in the bulletins of the UEPG—European Aggregates Association [5].

Among brick, glass, or soft materials—which are usually deposited at landfills—the most common waste is large size concrete debris. After the crushing process of this material is finished, concrete debris in a gradation curve from 0 mm to 63 mm are deposited and can be utilized as an aggregate or soil material. Commonly, in the literature, such porous media are called recycled concrete aggregate (RCA) and it is successfully used in road engineering. Therefore, a lot of geotechnical properties of RCA needed in road design are already known. For example, optimum moisture content for RCA with sandy gravel (saGr) gradation curve range from 8.35 to 11.74% [7,8]. Differences between optimum moisture content can occur due to various origins and different classes of concrete, from which the RCA was created. For example, RCA made of high-strength concrete with a lower water-to-cement ratio will present lower water absorption than RCA created from lower class concrete [6]. Another very important parameter in road engineering is California Bearing Ratio (CBR). Jiménez et al. [9] gives CBR values for RCA from 97 to 138%, Melbouci [10] gives one CBR value of 128%. According to Polish standards for roads WT-4 (Technical Specifications No. 4) [11], materials for the sub-base require a CBR value at least 60% and 80% for the base layer. Accordingly, RCA can be considered to be suitable for use in either sub-base or base layers. For correct prediction of road settlement resilient modulus is needed. Sas et al. [12] provides resilient modulus from 450 to 1710 kPa for numbers of loading cycles range from 10 to 50,000. They observed stiffness improvement with increase in numbers of loading cycles. Bozyurt et al. [13] presented empirical equations for resilient modulus based on particle shape, binder type and aggregate mineralogy of RCA. Arm (2001) in his cyclic triaxial tests show that RCA have the same resilient modulus as natural aggregate (NA), but over time laboratory and field tests showed an increase in stiffness for unbound layers with RCA. Arm [14] explained this fact by self-cementing properties of the unhydrated cement particle of RCA.

For other geotechnical applications, like aggregates for embankments or earth dams, mechanical parameters of RCA are needed. For RCA with gradation of saGr, O'Mahony [15] determined internal friction angle ($\varphi$) in direct shear test from 39.5 to 42°. Sas et al. [15] and Soból et al. [7] confirmed the test results presented by O'Mahony, but Sas et al. [16] from triaxial test estimated much higher value of $\varphi$ equal to 53°. The cohesion phenomena which was reported during shear strength tests under RCA is still not fully explained. Usually aggregates do not behave cohesion.

But RCA has a very complicated structure and self-cementing properties which can lead to cohesion in non-cohesion soil. However, from a geotechnical point of view, self-cementation process is a phenomenon which improves mechanical properties of RCA [14]. Earth construction mentioned earlier often deals with dynamic loads like weaving or vibration caused by cars or trains. Small strain dynamic parameters like shear wave velocity, shear modulus, and damping ratio are needed to predict behavior of this structures subjected to seismic loads. Sas et al. [17] describe different procedures to obtained shear wave velocities from bender elements test. Gabryś et al. [18] use described methodology for RCA and got shear wave velocities from about 175 m/s for effective stress 45 kPa to about 300 m/s for effective stress 180 kPa. He and Senetakis [19] also obtained a similar value of shear wave velocity. For determination maximum shear modulus, shear modulus degradation curve, and minimal damping ratio of RCA, resonant column apparatus in cyclic torsional shear mode was employed in Gabryś et al. [20] article. They obtained maximum shear modulus from 62 MPa at 45 kPa effective stress to 220 MPa at 225 kPa effective stress and minimal damping ratio from 0.5 to about 3%. These reports also confirmed He and Senetakis [19]. Moreover, RCA has higher shear modulus than NA at the same effective stress, because of its sharpness and roughness which causes chocking of grains.

The coefficient of permeability *k*, is a key parameter characterizing seepage in soils, which is one of the most important in design earth dams or levees. However, also in road design, filtration layers need to be especially well characterized by the *k* value. When constructing a dam or levees, material for its body should has high permeability coefficient. Nevertheless, also in road engineering, even the subbase should have a high *k* value. However, such tests are often abandoned because of the low budget of a given investment. General tests are replaced by simpler methods, such as predicting the hydraulic conductivity on the basis of the porosity or grain size distribution [21]. As Chapuis [21] reported, hydraulic conductivity depends on the pore sizes and the way in which they are connected. However, well-graded natural soils are characterized by the presence of rounded grains, mostly of quartz origin with small roughness. In the case of slight clay particles content, the pores are still large and well connected. In contrast, RCA is characterized by high roughness grains with irregular shapes, which complicates the flow paths of water. Moreover, the internal pores are very often present in RCA, which greatly increases the specific surface [22]. The values of the void ratio, porosity, and therefore hydraulic conductivity in case of RCA are higher than a natural aggregate. The porosity of RCA, reported by Gómez-Soberón [23], is 14.86% when natural aggregates have 3% porosity. The high porosity was found to be caused by the presence of cement mortar. Deshpande and Hiller [22] report differences between the characterization of aggregates during a comparison between natural and recycled aggregates. Water absorption in this study, made with respect to American Society for Testing and Materials, Standard Test Method for Relative Density (Specific Gravity) and Absorption of Coarse Aggregate: ASTM C 127, gave different results for RCA when helium pycnometer and image analysis were conducted. RCA is a granular material that, as opposed to natural aggregates, is coated with patches of cement mortar remains. The remains of cement paste impact the results of the experiment, designed originally for natural aggregates (Tam et al. 2008). RCA's water absorptions after a 24-hour test increased from 5.72% to 8.28%. It is worth noting that 80% of the total water absorption was reached after the first 5 h [24]. This difference compared to natural aggregates may impact the RCA permeability properties.

The hydraulic conductivity of RCA was studied in works whose main purpose was to obtain the geotechnical properties, and extensive studies in this subject need to be performed. Arulrajah et al. [25] found the coefficient of permeability $3.3 \times 10^{-8}$ m/s, which was similar to cohesive soils and materials used rather for dam and levee core than for dam body or filtration layer in road engineering. Nevertheless, a very low coefficient of permeability still was within the recommended value for a road subbase. Poon et al. [26] and Poon and Chan [27] reported the hydraulic conductivity of RCA in a range from $2.04 \times 10^{-3}$ to $2.67 \times 10^{-3}$ m/s. The differences between results [26,27] and [25] are very significant, which proves that more studies on this parameter should be carried out. The above-cited

works did not mention the existence of non-Darcian flow. Nevertheless, the results of the experiments conducted by the abovementioned authors are in range of $2.04 \times 10^{-3}$ to $3.3 \times 10^{-8}$ m/s, which can indicate the existence of non-Darcian flow. Hansbo [28] showed that consolidation process in the cohesive soils leads to the formulation of a theoretical approach for dealing with this problem. The water flux in this study is proportional to a power function of the hydraulic gradient if the tested gradient is lower than a certain critical value. After this value, the gradient becomes linear for large gradient values. This behavior was explained by assuming that there is a certain hydraulic gradient in clayey soils that causes the binding energy to overcome the energy of mobile pore water. The threshold gradient is one of the parameters that constitute Hansbo's non-Darcian flow.

RCA formed on the basis of the different concrete will have unique chemical properties along with the different raw materials used in its production. As a consequence, it is not possible to identify all sources of heavy metals, but it is possible to identify materials that may be their source. Such materials include cement. The properties of heavy metals influencing their presence in the production process are their volatility. Non-volatile compounds—i.e., Ba, Be, Cr, As, Ni, V, Al, Ti, Ca, Fe, Mn, Cu, and Ag—leave the furnace as cement clinker components and are the source of metal content in the cement composition and thus also the source of heavy metals in RCA. In Table 1, the chemical composition of the basic constituents of the concrete from which the RCA is produced is presented. Presented data concerns only one kind of concrete and is not a general description for such material.

**Table 1.** Chemical composition of raw materials used in the production of concrete [29].

| Parameter | $SiO_2$ (%) | $Al_2O_3$ (%) | $Fe_2O_3$ (%) | $TiO_2$ (%) | CaO (%) | MgO (%) | MnO (%) | $SO_3$ (%) | $K_2O$ (%) | $Na_2O$ (%) | $P_2O_5$ (%) |
|---|---|---|---|---|---|---|---|---|---|---|---|
| Cement | 13.95 | 5.35 | 4.88 | - | 61.44 | 1.20 | 0.55 | 2.95 | 0.78 | 0.22 | 0.1717 |
| Fly ash | 50.40 | 27.31 | 4.79 | 1.50 | 7.29 | 1.49 | 0.06 | 0.46 | 1.52 | 0.28 | 1.06 |
| Sand | 26.66 | 1.76 | 1.00 | 0.09 | 30.85 | 6.89 | 0.04 | 0.01 | 0.24 | 0.22 | 0.00 |
| Gravel | 14.34 | 1.31 | 0.74 | 0.07 | 36.24 | 8.59 | 0.03 | 0.01 | 0.10 | 0.14 | 0.0 |

The high content of heavy metals and sulphates can lead to leaching of these compounds during a filtration process, which can cause significant environmental pollution. Therefore, it is necessary to analyze these RCA properties. One of the researchers that conduct studies of the leaching properties of RCA was Barbudo et al. [30]. They examined 17 mixtures of recycled aggregates with different contents of RCA, NA, ceramic, gypsum, bituminous, and other substances in laboratory. Barbudo et al. [30] proved that different mixtures with RCA do not exceeds the permitted standards of concentrations of heavy metals according to European Union (EU) Landfill Directive [31]. Similar study to [30] was performed by Engelsen et al. [32], but they conducted their tests in the field. They use RCA to build sub-base layers in three section on highway No. E6 located 20 km south from Oslo. Outgoing water from each sections of the highway was collected and tested for four years between 2006 and 2010. Concentration of heavy metals exceeds the limit only in the case of chrome. However, it could be connected with petroleum pollution from passing vehicles.

For successful application of RCA in road engineering and as an aggregate for earth dam, levees, and embankments, the permeability of RCA needs to be examined and analyzed. The aim of this study was to evaluate the use of aggregate from recycled concrete in road and embankments construction in terms of its permeability based on the designation of the coefficient of permeability and its variation, which was observed during this study. The main objective of the study was to determine the threshold gradient. Therefore, in the first instance, a series of analyses has been made in order to verify the correctness of the tests. The errors arising from the previous assumptions have also been estimated. Moreover, laboratory leaching tests of RCA was performed to determine potential environmental pollution.

## 3. Materials and Methods

### 3.1. Material

In this research, a tested demolished concrete was taken from a building demolition site by the skid-mounted impact crusher. The strength class properties of the construction concrete, made from Portland cement, were estimated to be from C16/20 to C30/35 based on the data obtained from building plans. The obtained material was later sieved into appropriate fractions with application of the [33]. The RCA was divided into three groups, namely, blends 1, 2, and 3. Each blend was composed from sieved fractions. The aggregates were 99% composed from broken cement concrete, the rest being glass and brick ($\Sigma$(Rb, Rg, X) $\leq$ 1% m/m), in accordance with standard [34,35], and contain no asphalt or tar elements. A grain gradation curve was adopted in accordance with the Polish technical standard [11], which is a common technological guideline for road engineers, and placed between the upper and lower grain gradation limits. Besides, the created mixture is appropriate for earth structures like dams or embankments.

In order to estimate the physical properties, a series of tests were conducted. The sieve analysis led to classifying the material as sandy gravel (saGr), in accordance with Eurocode 7 [36]. The test results are shown in Figure 1. This distribution of particles from 16 mm to 0 mm is in range of the standard for aggregates used as auxiliary subbase and improved subgrade in road engineering and in earth structures according to [11]. The coefficient of uniformity, $C_U$, indicates roasting granularity for all blends, but for blend 1, this value is significantly lower, mostly because of a low maximum grain diameter when compared to the other two blends. The coefficient of curvature, from 1.81 for blend 1, 2.16 for blend 3, and 5.56 for blend 2, indicates that this material is grading well. Moreover, the high value of $C_U$ and $C_C$ testify of the good compactility [37] of the tested RCA, which is very important in earth construction. During this study, the void ratio for the 0–8 mm blend was approximately 0.386, for the blend 0–16 mm was approximately 0.543 and for the 0.05–16 mm blend was approximately 0.656.

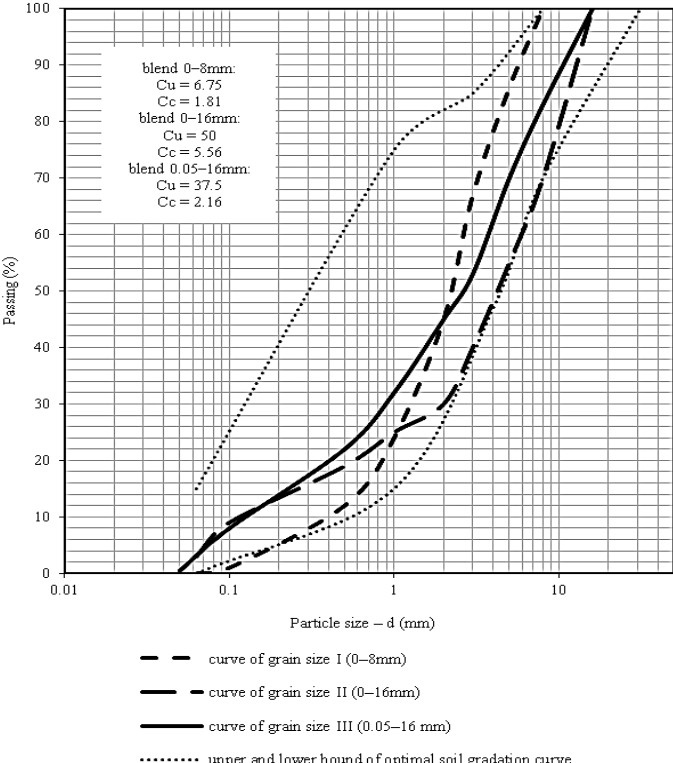

**Figure 1.** Grain size distribution curve of each blend.

### 3.2. Permeability Test

There exist a variety of methods of permeability measurements using field methods that utilize BAT piezometers [38] and the falling head method with a flow pump [39–41]. However, a constant head method device was used in this study for estimating the basic characteristics of RCA because of its simplicity and invariable conditions of test. Moreover, the constant head method is one of the most reliable techniques of permeability measurements.

In this paper for permeability testing, laboratory equipment was used. A constant head device (called in this study as permeameter) is shown in Figure 2. Permeameter construction consists of inner and outer cylinders made form stainless steel (inner cylinder dimensions: height h = 0.17 m, diameter d = 0.205 m, outer cylinder dimensions: h = 0.27 m, d = 0.19 m), which are connected by a permeability mold also made from stainless steel, where the sample is placed. The permeability mold has a perforated bottom. After placing a sample, a perforated cover is installed on top. The permeability mold is fixed to the inner cylinder by four screws and an o–ring to make sure that no unexpected seepage of water from outer cylinder occurs. The principle on which this device operates relies on communicating vessels, which allow the water to flow from the outer cylinder to the inner cylinder through the soil sample. The hydraulic gradient is simply set by the difference between the outer and inner water table heights. In practice, the inner water table is fixed, and the hydraulic gradient is inflicted by a changeable outer water table height. Tests were conducted when both the inner and outer water tables were in the fixed position. Measurements of the outflow water in time were repeated five times for each test point.

Tests were performed on three blends of grain, whose size was 0–8, 0–16, and 0.05–16 mm. The sample presented in Figure 1 were compacted in a permeability mold using Proctor's method. The energy of compaction was 0.59 J/cm3, and the mass of the compaction hammer was 2.5 kg. Compaction was conducted in three layers, where 16 strokes were performed for each layer. The sample volume was $6.34 \times 10^{-4}$ m$^3$, with a diameter of 0.116 m and a height of 0.06 m. Proctor's method of compaction was used for the proper simulation of conditions that are present in the compacted layers of a road or embankment. Nevertheless, before compaction in a permeameter mold, a proper Proctor's study has been conducted. The RCA's maximal density and optimal moisture content were estimated as 1960 kg/m$^3$ and 8.0%, respectively.

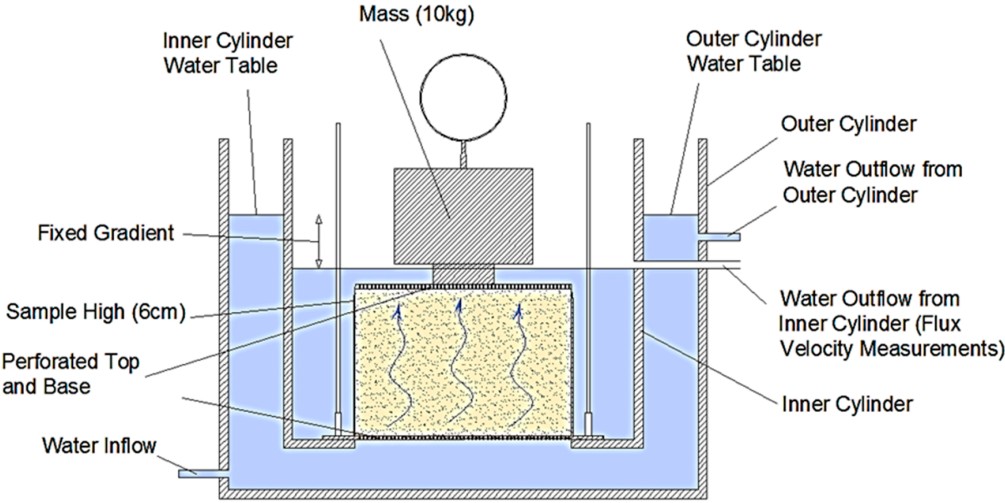

**Figure 2.** Permeameter scheme.

The procedure of the compaction and testing of the sample was upgraded several times before the main study in order to eliminate common mistakes, as reported by Chapuis [21].

(1) Compaction was performed with respect to the Proctor method in order to avoid crushing the grains, and this operation was performed in the permeability mold because of the risk of creating voids around the sample and the rigid wall. Compaction of samples was performed in the optimal moisture content, which was 8% [41]. This action was vital in order to simulate the performance of the subbase soil layer or compaction process in earthen construction.

(2) It is obvious that the permeability tests' saturation degree, $S_r$, has to be equal to 1. To solve this issue, after its installation on the permeameter, the sample was filled by aerated water at a constant rate of 0.25 mm/h of the water table, which, for a 6 cm sample, takes 24 h.

(3) Construction of the permeameter [41] made it possible to avoid hydraulic head loss because of the constant inner and outer water tables. The outflow was measured exactly next to the outflow pipe, which, during the tests, was never fully filled.

(4) Moreover, the movement of fine particles was controlled by checking the inside of the cylinder after the tests. Weight installed on the top of the perforated cover prevented the soil skeleton from movement due to the seepage of water.

(5) The pressure on the soil skeleton was equal to 10 kPa. The permeability coefficient was tested using the hydraulic gradients typical for water damming construction, which were 0.2, 0.3, 0.4, 0.5, 0.58, 0.67, 0.75, and 0.83, performed after 40 trials at each of the hydraulic gradients for each of the blends.

### 3.3. Chemical Analysis

Analysis of concentrations of chlorides, sulphates, and heavy metals in blend 1, 0–16 mm was studied. Samples for examines concentration of sulphates and chlorides was prepared according to Eurocode 7 standards (PN-EN 1744-1:1998) [42]. For determination of water-soluble sulphates and chlorides, methodology from Kiedryńska et al. [43] was employed. Namely, for chlorides concentration Mohr method and for sulphates concentration turbidimetric method were used. In the case of heavy metals concentrations, samples were prepared according to PN-Z-15009:1997 [44] standards. Atomic absorption spectrometry was use for identify concentrations of Co., Cd, Cu, Ni, Pb, and Zn in prepared specimens. Measurements was carried out on ASA ICE 3000 Series AA Spectrometer (produced by: Thermo Scientific, Waltham, MA, USA) device. Moreover, content of heavy metals in one kilogram of dry waste was calculated. Electrolytic conductivity and pH was also estimated. Every tests were repeated three times. During the all tests room temperature was fixed at 25 °C.

## 4. Results

### 4.1. Threshold Gradient and Suffosion

The permeability in soils obeys the Darcian law. The flux velocity $v$ which is velocity the rate of flow of water is calculated as a product of coefficient of permeability k and hydraulic gradient i. The hydraulic gradient is a head difference inversely proportional to length. The studies under permeability of soils lead to find that not all types of soil behave Darcy's law. Figure 3 presents the relationships between the hydraulic gradient and the flow velocity in non–Darcian flow, proposed by Hansbo [28]. The RCA tests results have shown existence of non–Darcian flow. For engineering applications, non–Darcian flow is simplified to Darcian flow when the seepage of water starts from a certain gradient, which is called the threshold gradient. The existence of the threshold gradient comes from the assumption that when the gradient value is less than the threshold gradient, the flow rate may dramatically decrease, which follows non–Darcian flow [28].

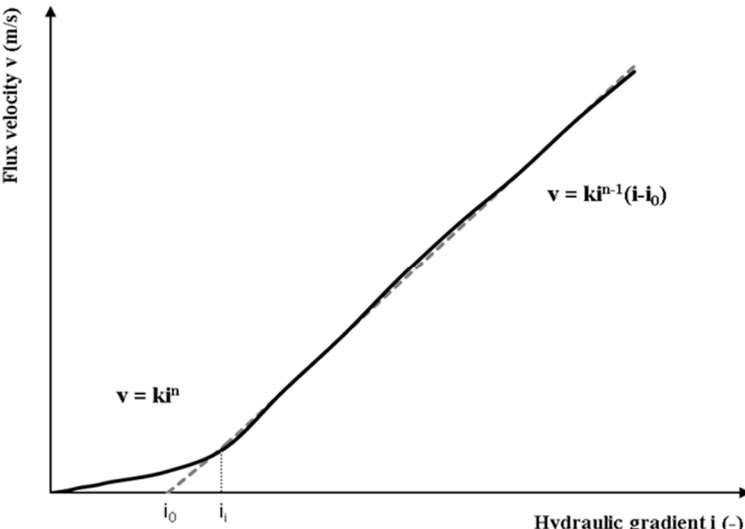

**Figure 3.** The relationship between the flux velocity and the hydraulic gradient in non-Darcian flow [28].

To exclude the possibility of unexpected suffosion during the tests, which could disrupt collected data, the occurrence of suffosion by Kenney and Lau [45–48] was analyzed. The results of the above analysis are presented in Figure 4. It has been concluded that suffosion occurs only in the blend whose particle size is 0.05–16 mm and only at a higher grain diameter, which is indicated by the occurrence of the crushing of the sample during the test. Nevertheless, the top perforated cover does not allow the grains to escape from the filtration hoop.

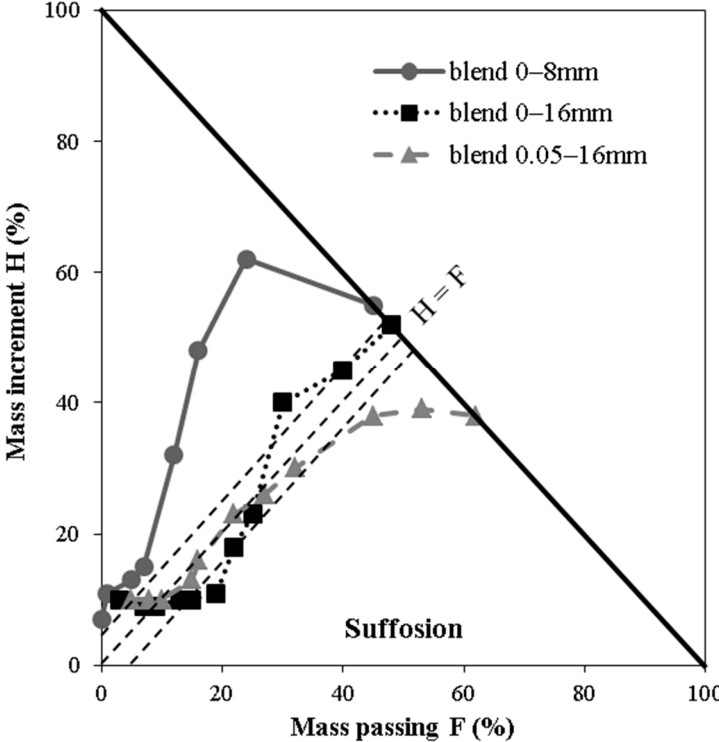

**Figure 4.** Results of suffosion analysis in respect to Kenney and Lau method [45] for the three blends of the RCA.

### 4.2. Results of the Permeability Tests

Figure 5 presents the averaged results of the flux velocity for each gradient. In the graphs, two phases of seepage can be distinguished. The first pre-linear stage provides a low permeability in the blends with small gradients (i = 0.2–0.3). The second phase characterizes the linear flow in accordance with Darcy's law. For both phases of the flow, the $R^2$ value has been calculated. For the first phase of the pre-linear stage, the results were in the range of $R^2$ equal to 0.9853 to 0.9997. For the second phase, $R^2$ ranged from 0.9801 to 0.9966. The distribution of test results consists of the breakdown phase of the pre-linear and linear phases and is consistent with the theoretical recognition presented on Figure 3.

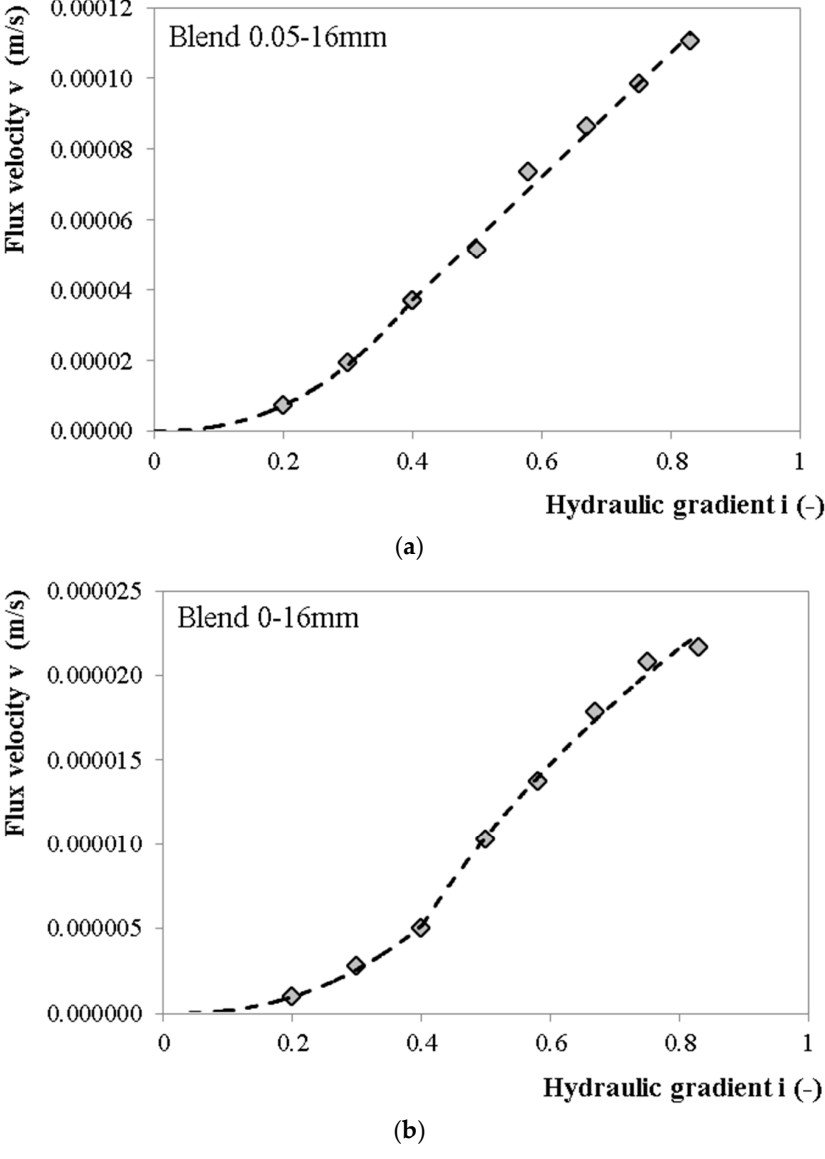

(a)

(b)

**Figure 5.** *Cont.*

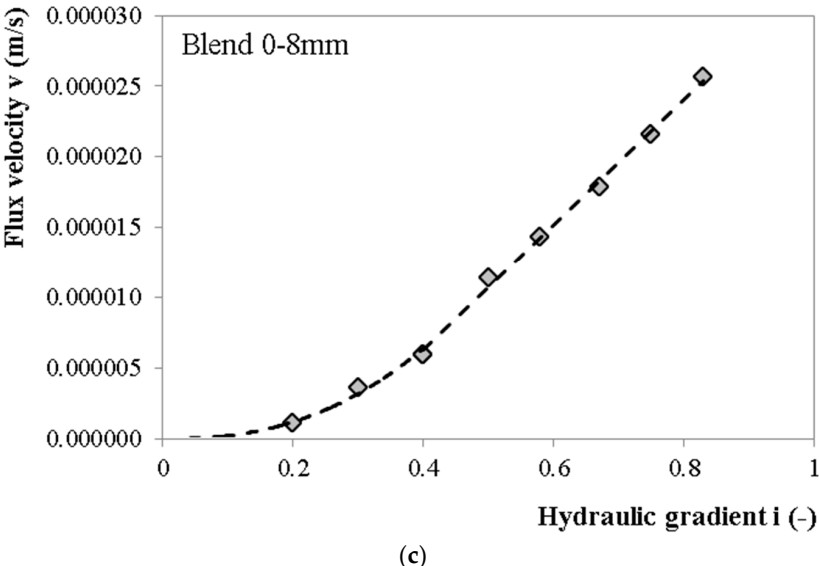

(c)

**Figure 5.** The relationship between the flux velocity and the hydraulic gradient for the blends. (**a**) blend 0.05–16 mm; (**b**) blend 0–16 mm; (**c**) blend 0–8 mm.

The results of the tests shown in Figure 5 clearly indicate the existence of the threshold gradient. For low gradients (0.1–0.3), the flux velocity, $v$, is equal $v = k \cdot i^n$, where k is the coefficient of permeability, i is the gradient value and n is a parameter, which, in case of this test, was equal to 1.6 for the 0.05–16 mm blend. For the other two blends, n was equal 1.4.

### 4.3. Statistical Reliability of the Permeability Tests Results

In order to verify the normality, a Q–Q (quantile-quantile) plot graph and the Shapiro–Wilk test were used. The null hypothesis ($H_0$) for the Shapiro–Wilk test is that the distribution is normal for the result with $p \geq 0.05$. In the case of $p < 0.05$, the null hypothesis was rejected. The results for the Shapiro–Wilk test for all blends estimate the distribution for all blends as a normal distribution. Figure 6 shows the graphs for the Q–Q plot.

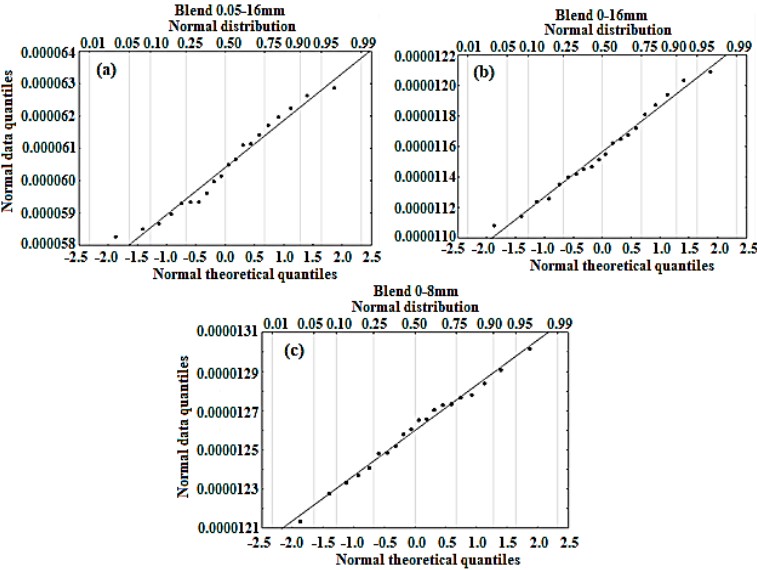

**Figure 6.** The Q–Q (quantile-quantile) plot graph for the blends. (**a**) blend 0.05–16 mm; (**b**) blend 0–16 mm; (**c**) blend 0–8 mm.

For a better illustration of the nature and structure of the results and the accuracy of the tests performed, Table 2 summarizes the results of the analysis of errors for each of the tested gradients.

**Table 2.** Estimated errors of the flux velocity measurements for each of the gradients and blends.

| Hydraulic Gradient i (–) | Standard Deviation | Absolute Error of Measurement (m/s) | Relative Error (m/s) | Percent of Error (%) |
|---|---|---|---|---|
| | | **Blend 0.5–16 mm** | | |
| 0.2 | $3.4 \times 10^{-6}$ | $1.0 \times 10^{-8}$ | $1.1 \times 10^{-4}$ | $7.0 \times 10^{-4}$ |
| 0.3 | $3.5 \times 10^{-6}$ | $-4.0 \times 10^{-8}$ | $-2.3 \times 10^{-4}$ | $1.9 \times 10^{-3}$ |
| 0.4 | $2.4 \times 10^{-6}$ | $-1.0 \times 10^{-8}$ | $-4.0 \times 10^{-4}$ | $3.7 \times 10^{-3}$ |
| 0.5 | $1.4 \times 10^{-7}$ | $1.0 \times 10^{-8}$ | $1.0 \times 10^{-4}$ | $5.1 \times 10^{-3}$ |
| 0.58 | $1.5 \times 10^{-7}$ | $-1.0 \times 10^{-8}$ | $-2.0 \times 10^{-4}$ | $7.4 \times 10^{-3}$ |
| 0.67 | $5.8 \times 10^{-7}$ | $-1.0 \times 10^{-8}$ | $-2.0 \times 10^{-4}$ | $8.6 \times 10^{-3}$ |
| 0.75 | $1.2 \times 10^{-6}$ | $9.0 \times 10^{-8}$ | $9.0 \times 10^{-4}$ | $9.8 \times 10^{-3}$ |
| 0.83 | $1.8 \times 10^{-6}$ | $-1.2 \times 10^{-7}$ | $-1.0 \times 10^{-3}$ | $1.11 \times 10^{-3}$ |
| | | **Blend 0–16 mm** | | |
| 0.2 | $5.4 \times 10^{-7}$ | $3.0 \times 10^{-8}$ | $3.1 \times 10^{-2}$ | $1.0 \times 10^{-4}$ |
| 0.3 | $5.4 \times 10^{-7}$ | $3.0 \times 10^{-8}$ | $1.3 \times 10^{-2}$ | $3.0 \times 10^{-4}$ |
| 0.4 | $2.0 \times 10^{-7}$ | $1.0 \times 10^{-8}$ | $2.2 \times 10^{-3}$ | $5.0 \times 10^{-4}$ |
| 0.5 | $3.7 \times 10^{-7}$ | $-2.0 \times 10^{-8}$ | $-1.6 \times 10^{-3}$ | $1.0 \times 10^{-3}$ |
| 0.58 | $6.0 \times 10^{-7}$ | $3.0 \times 10^{-8}$ | $2.4 \times 10^{-3}$ | $1.4 \times 10^{-3}$ |
| 0.67 | $4.1 \times 10^{-7}$ | $3.0 \times 10^{-8}$ | $1.7 \times 10^{-3}$ | $1.8 \times 10^{-3}$ |
| 0.75 | $1.9 \times 10^{-7}$ | $0$ | $1.0 \times 10^{-4}$ | $2.1 \times 10^{-3}$ |
| 0.83 | $2.7 \times 10^{-7}$ | $2.0 \times 10^{-8}$ | $9.0 \times 10^{-4}$ | $2.2 \times 10^{-3}$ |
| | | **Blend 0–8 mm** | | |
| 0.2 | $5.4 \times 10^{-7}$ | $4.0 \times 10^{-8}$ | $3.9 \times 10^{-2}$ | $1.0 \times 10^{-4}$ |
| 0.3 | $5.7 \times 10^{-7}$ | $4.0 \times 10^{-8}$ | $1.2 \times 10^{-2}$ | $4.0 \times 10^{-4}$ |
| 0.4 | $1.6 \times 10^{-7}$ | $1.0 \times 10^{-8}$ | $2.5 \times 10^{-3}$ | $6.0 \times 10^{-4}$ |
| 0.5 | $2.4 \times 10^{-7}$ | $2.0 \times 10^{-8}$ | $1.3 \times 10^{-3}$ | $1.1 \times 10^{-3}$ |
| 0.58 | $2.6 \times 10^{-7}$ | $-1.0 \times 10^{-8}$ | $-4.0 \times 10^{-4}$ | $1.4 \times 10^{-3}$ |
| 0.67 | $4.7 \times 10^{-7}$ | $0$ | $2.0 \times 10^{-4}$ | $1.8 \times 10^{-3}$ |
| 0.75 | $2.9 \times 10^{-7}$ | $3.0 \times 10^{-8}$ | $1.3 \times 10^{-3}$ | $2.2 \times 10^{-3}$ |
| 0.83 | $5.0 \times 10^{-7}$ | $5.0 \times 10^{-8}$ | $2.1 \times 10^{-3}$ | $2.6 \times 10^{-3}$ |

The analysis of the test results in terms of the standard deviation and the interval between the highest and lowest result flux velocity for tests are also included (Figure 7). For the research conducted at low gradients of 0.2–0.3, the values for both the standard deviation and the interval were significantly higher. On this basis, we can conclude that the study at these gradients is unstable.

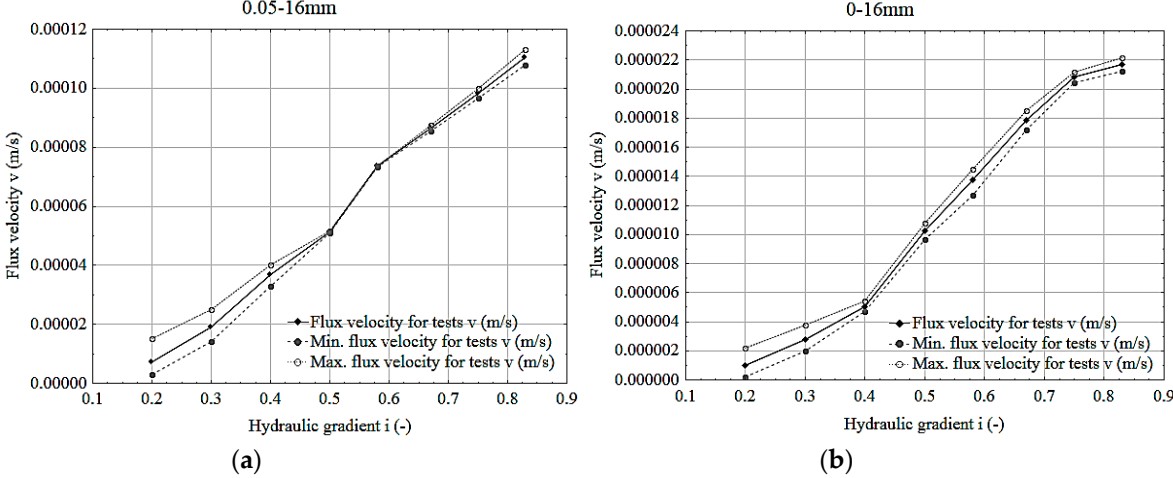

**Figure 7.** *Cont.*

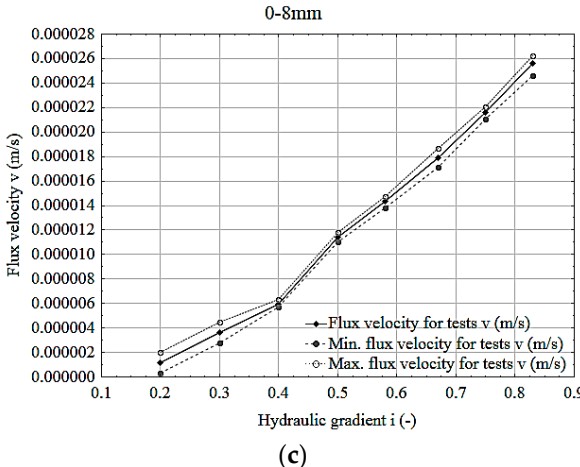

(c)

**Figure 7.** Prediction of the threshold gradient on the basis of the characteristic linear trend line with the minimal and maximal value flux velocities. (**a**) blend 0.05–16 mm; (**b**) blend 0–16 mm; (**c**) blend 0–8 mm.

In order to determine the estimated value before calculating the flux velocity average, all coefficients have been registered during the laboratory filtration process. The coefficient of permeability for all tests results was the mean, in accordance with which the average value of coefficient of permeability was calculated and presented in Table 3.

Next, the flux velocity was calculated using Darcy's Law, $v = k \cdot i$, where $v$ is the flux velocity (m/s), k is the coefficient of permeability (m/s), and i is the hydraulic gradient, also included in Table 4. In the remainder of the paper, the results obtained on the basis of these calculations will have a specific name, flux velocity for $k_{avg}$.

In the next step, a graph of the relationship between the average of the flux velocity and the hydraulic gradient was constructed. The trend line was derived for these values, taking into account the confidence interval at 0.95 to define the threshold gradient. The test results on unstable gradients 0.2 and 0.3 were derived with an additional trend line, marked on Figure 8, representing an unstable flux velocity, in order to verify whether this will cause the cutting of the *x*-axis at 0 of the confidence interval for the used linear trend line equation. It can be concluded from Figure 8 that the points of intersection with the *x*–axis for the unstable flux velocity are within the confidence interval or at the border, designated for all tests at specified gradients for each blend.

**Table 3.** Calculation of the flux velocity on the basis of the mean coefficient of permeability.

| i (–) | Blend 0.05–16 mm | | Blend 0–16 mm | | Blend 0–8 mm | |
|---|---|---|---|---|---|---|
| | $k_{avg}$ (m/s) | $v$ (m/s) | $k_{avg}$ (m/s) | $v$ (m/s) | $k_{avg}$ (m/s) | $v$ (m/s) |
| 0.2 | | $2.0 \times 10^{-5}$ | | $3.8 \times 10^{-6}$ | | $4.2 \times 10^{-6}$ |
| 0.3 | | $3.1 \times 10^{-5}$ | | $5.7 \times 10^{-6}$ | | $6.2 \times 10^{-6}$ |
| 0.4 | | $4.1 \times 10^{-5}$ | | $7.6 \times 10^{-6}$ | | $8.3 \times 10^{-6}$ |
| 0.5 | $1.0 \times 10^{-4}$ | $5.1 \times 10^{-5}$ | $1.9 \times 10^{-5}$ | $9.5 \times 10^{-6}$ | $2.1 \times 10^{-5}$ | $1.0 \times 10^{-5}$ |
| 0.58 | | $5.9 \times 10^{-5}$ | | $1.1 \times 10^{-5}$ | | $1.2 \times 10^{-5}$ |
| 0.67 | | $6.8 \times 10^{-5}$ | | $1.3 \times 10^{-5}$ | | $1.4 \times 10^{-5}$ |
| 0.75 | | $7.6 \times 10^{-5}$ | | $1.4 \times 10^{-5}$ | | $1.6 \times 10^{-5}$ |
| 0.83 | | $8.5 \times 10^{-5}$ | | $1.6 \times 10^{-5}$ | | $1.7 \times 10^{-5}$ |

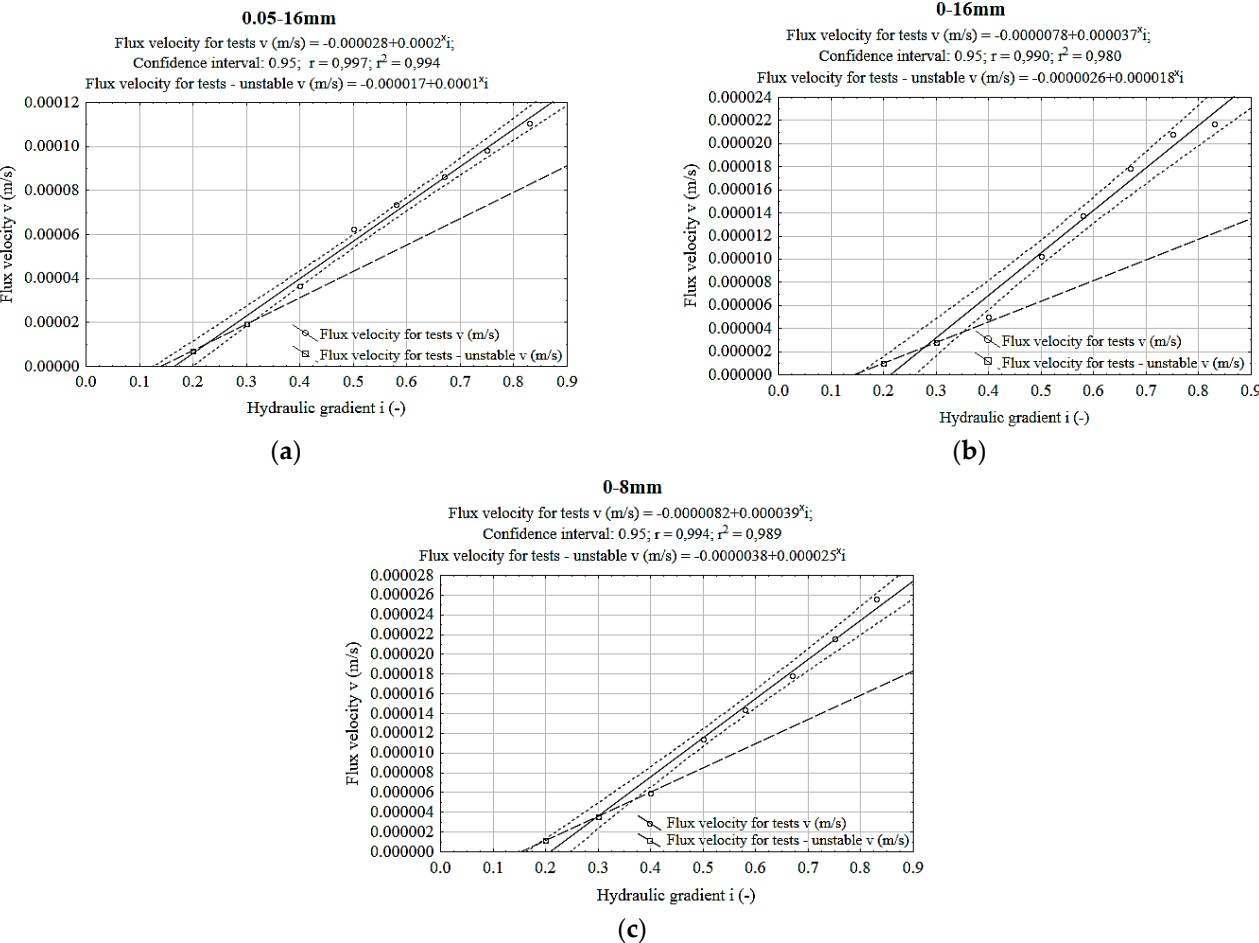

**Figure 8.** Relation between the flux velocity and the hydraulic gradient with unstable tests for the blends. (**a**) blend 0.05–16 mm; (**b**) blend 0–16 mm; (**c**) blend 0–8 mm.

Figure 9 presents the two methods of determining the threshold gradient ($i_0$). The first is a simplified method based on $k_{avg}$ and on the basis of the determination of the flux velocity described earlier. The calculations are shown in Table 4. The second method involves the determination linear trend line for the average of the test results for each gradient.

**Table 4.** Summary of the calculation results for the threshold gradient calculation.

| | |
|---|---|
| **Blend 0.05–16 mm** | $v = -0.000029 + 0.0002i$<br>Threshold gradient = 0.175<br>Confidence interval (0.95) — P (0.145 < 0.175 < 0.205) |
| **Blend 0–16 mm** | $v = -0.0000078 + 0.000037i$<br>Threshold gradient = 0.212<br>Confidence interval (0.95) — P (0.150 < 0.212 < 0.262) |
| **Blend 0–8 mm** | $v = -0.0000082 + 0.000039i$<br>Threshold gradient = 0.210<br>Confidence interval (0.95) — P (0.165 < 0.210 < 0.250) |

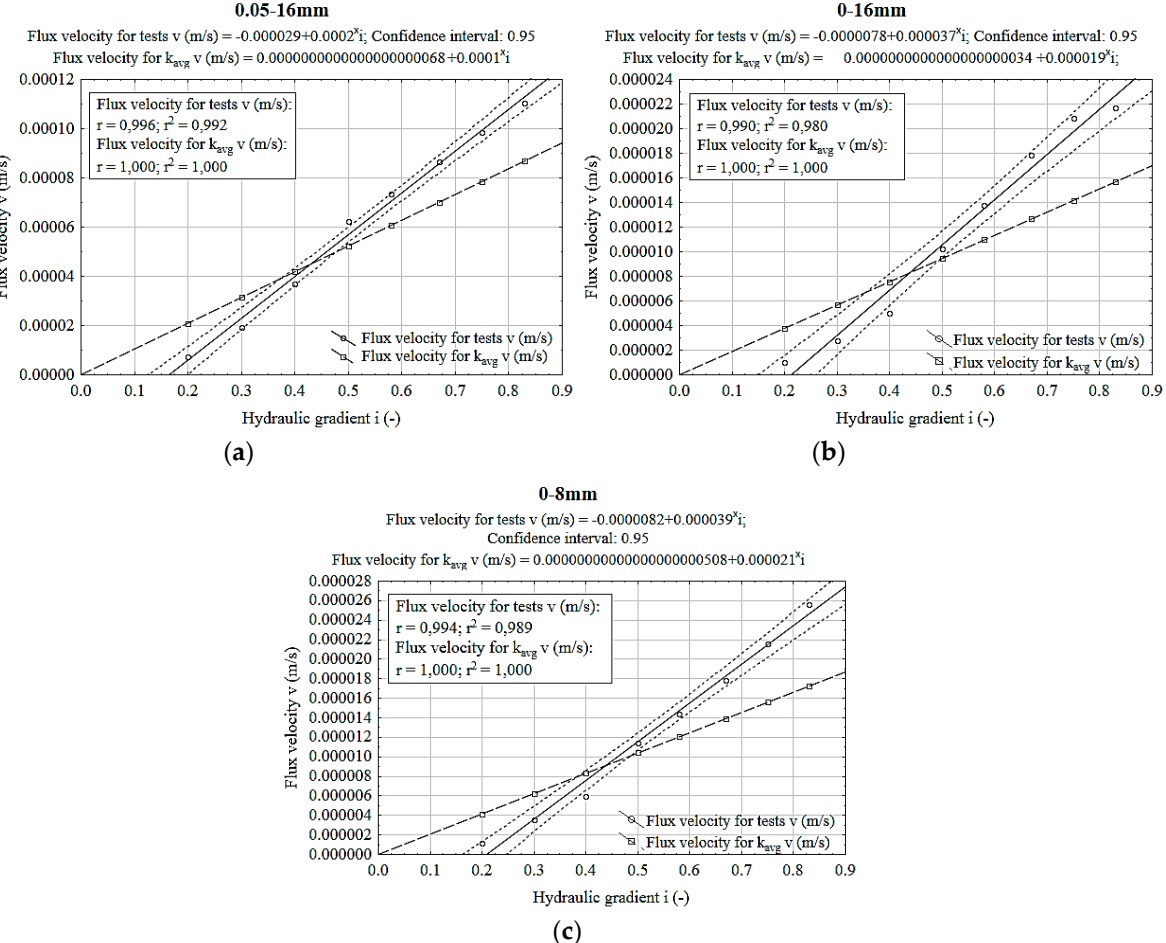

**Figure 9.** Prediction of the threshold gradient on the basis characteristic linear trend line for blend 0–8 mm. (**a**) blend 0.05–16 mm; (**b**) blend 0–16 mm; (**c**) blend 0–8 mm.

*4.4. Chemical Analysis Tests Results*

Results from conducted leaching tests of RCA are presented in Table 5. Every value from the Table 5 is the average from three performed tests. It can be seen that none of concentration values do not exceed acceptance criteria. However, concentration of cobalt and cadmium are well below the limit, which is 1.0 in case of cobalt and 0.05 in case of cadmium. Concentrations of copper and nickel are higher than metals mentioned earlier, but still about four times below the limits. Various values of concentration of these metals can be found in the literature. Galvin et al. [49] gives the values of copper and nickel equal to 0.28 and 1.75 mg/kg, but Del Rey et al. [50] provides 0.01 and 0.01 mg/kg. Lead and zinc have been not found in water extract made from RCA in this study, which is the confirmation of the Barbudo et al. [30] research. Concentration of sulphates is equal to 194.7 mg/L, which are the nearest to the acceptance criteria. Similar values of sulphate concentrations are reported in [31,50]. Chlorides concentration are on safe level of 14.05 mg/L in presented research. Galvin et al. [49] and Del Rey [50] reported higher chlorides concentration but still meet the norm. Moreover, Del Rey et al. [50] provided value of electrolytic conductivity 850 µS/cm, which is also higher than C = 501.5 µS/cm obtained in this study. However, both values indicated on little pollution of water extract made from RCA. Additionally, leaching of heavy metals are limited due to alkaline pH of RCA water solution. Moreover, small concentration of heavy metals are connected with concrete properties, which binds heavy metal compounds during the hardening process of concrete.

**Table 5.** Leachate concentration from recycled concrete aggregate.

| Element | Co. (mg/L) | Cd (mg/L) | Cu (mg/L) | Ni (mg/L) | Pb (mg/L) | Zn (mg/L) | Sulphates (mg/L) | Chlorides (mg/L) | C (µS/cm) | pH |
|---------|-----------|-----------|-----------|-----------|-----------|-----------|------------------|------------------|----------|-----|
| Value | 0.066 | 0.00067 | 0.121 | 0.127 | n. d. | n. d. | 194.7 | 14.05 | 501.5 | 7.91 |
| Acceptance criteria * | 1.0 | 0.05 | 0.5 | 0.5 | 0.5 | 2.0 | 500 | 1000 | - | - |

* Official Gazette of the Republic of Poland, Regulation of the Minister of the Environment of 18 November 2014 on the conditions to be met for the introduction of sewage into waters or to land and on substances particularly harmful to the aquatic environment.

## 5. Conclusions

In this paper, recycled concrete aggregates (RCA) were characterized using a permeameter with an upgraded constant head method in order to avoid the common errors encountered in such methods. The results were statistically analyzed to estimate the statistical tolerance. The suffosion and coefficient of permeability were also calculated. Moreover, leaching properties of RCA was analyzed in order to identify potential environmental threats connected with filtration process. The conclusions are summarized below:

1. RCA exhibit the non-Darcian flow of water with threshold gradient occurrence.
2. The value of the coefficient of permeability, k, changes with the void ratio, e, exponentially, and during further studies, empirical equations were determined.
3. The flow of water trough RCA is very sensitive, and in the case of one blend, turbulent flow was observed around a critical gradient, which, for this material, was 0.9.
4. RCA proved its good quality as a permeable material, which is characterized by a coefficient of permeability of 0.05–16, 0–16, and 0–8 mm, and the k values were $1.018 \times 10^{-4}$, $1.89 \times 10^{-5}$, and $2.08 \times 10^{-5}$ m/s, respectively.
5. The threshold gradient was estimated in all blends, and statistical analysis shows the dependence of this phenomena on the fine particle content.
6. Fines also seem to be the reason behind the differences in the flux velocity between blends which do and do not contain them.
7. The threshold gradients for the tested blends of 0.05–16, 0–16, and 0–8 mm were 0.175, 0.212, and 0.210, respectively. Below theses gradients, for the tested blends the flux velocity may dramatically decrease, and water that stays in the RCA can degrade its mechanical parameters, including the bearing capacity, when road construction is considered or slope stability when earth dam or embankment construction is considered.
8. For road construction with standard gradients between 0.3 to 0.6, RCA has a constant value. Nevertheless, the threshold gradient needs to be taken into consideration when a large amount of fine particles is present in the material.
9. Obtained permeability coefficient for all examined blends are appropriate and meet requirements of aggregate for body construction of earth dams, levees, or embankments.
10. For earth dams or levees, a construction blend of 0.05–16 mm requires a reverse filter to avoid suffosion during filtration process.
11. Leaching of heavy metals during filtration process does not exceed the permitted limit. Although, the concentration of compounds having hexavalent chrome might exceed the limit. From a chemical point of view, RCA might be used as filtration layers in road, earth dam, or levee construction. Nevertheless, the concentration of other compounds harmful for life have to be checked before application.

The above results suggest the existence of possible permeability problems in roads or dams containing RCA layers. Nevertheless, such an occurrence concerns low gradients (below 0.3) and should be not an eliminative factor for application in above mentioned constructions.

**Author Contributions:** A.S. and W.S. conceived and designed the experiments; E.S. and A.G. performed the experiments; A.G., J.D., and E.S. analyzed the data and wrote the paper; W.S. and A.S. edited and audited the content.

**Funding:** This research received no external funding.

**Conflicts of Interest:** The authors declare no conflict of interest.

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
