# Peer review of "Permeability and Leaching Properties of Recycled Concrete Aggregate as an Emerging Material in Civil Engineering"

_applsci, doi:10.3390/app9010081_

Reviewer 1 Report

Permeability test was conducted in a certain range from 0-16mm. However, subbase materials at least under 40mm are to be normally used. Thus entire gradation analysis may be needed for RCA materials. Lines 195-196; For subbase materials for pavements, their gradation range was mentioned to be 0-16mm, saying that “This distribution of particles from 16mm to 0 mm is the standard for aggregates used s subbase in road engineering and in earth structures.

Lines 184-186; Out of C16/20 to C30/35 old concrete, how would you produce blends 1,2 and 3 RCA materials? Was the maximum size of RCA 16mm? Authors need to refer any gradation test standards adopted in this test.

Authors did not explain in detail what test procedures has been altered.

Authors may check hexavalent chrome limitation required for dam application. Authors explained this in lines 170-171, saying that “Concentration of heavy metals exceeds the limit only in the case chrome.” If this RCA sample does exceed the limit, it must not be used for the dam application.

If it does not follow the Darcian rule, I guess we may not use the constant head method.

Table 1. Chemical composition of raw materials used in the production of concrete. I am not sure whether this referred data is typical for a normal weight concrete. If this is one of any data, authors may explain this in the text. In addition, RCA needs to be included in Table 1.

Unnecessary concluding remarks need to be removed, at least i.e. 12, 14,…

[Minor and English check]

Lines 37; what does active composting mean?

Lines 56-57; “Designers and contractors are afraid of use recycled aggregates, because their attached and aware of the natural aggregate behavior.” Please rewrite this.

Lines 90-91; “Researchers still not fully explained cohesion phenomena which sometimes appears in RCA.” Rewrite this.

Limes 179; “ … in order to examined potential environmental pollution resulting from use the RCA in mentioned construction, laboratory leaching properties of RCA was determined.” Please rewrite this.

Author Response

Respond to Review Report 1 for submission of a paper to Applied Sciences MDPI journal

Andrzej Głuchowski, Wojciech Sas, Justyna Dzięcioł, Emil Soból, Alojzy Szymański
Warsaw University of Life Sciences, Faculty of Civil and Environmental Engineering
166 Nowoursynowska str. 02787 Warsaw, Poland

20.11.2018

Dear Reviewer,

We wish to thank You for all remarks. We carefully have read your review report and we prepared answers to your comments. The respond to this review ate accompanied by appropriate changes in the manuscript according to Your comments.

Permeability test was conducted in a certain range from 0-16mm. However, subbase materials at least under 40mm are to be normally used. Thus entire gradation analysis may be needed for RCA materials. Lines 195-196; For subbase materials for pavements, their gradation range was mentioned to be 0-16mm, saying that “This distribution of particles from 16mm to 0 mm is the standard for aggregates used s subbase in road engineering and in earth structures.

The gradation which was applied in this study is in range of the Polish standards for auxiliary subbase and improved subgrade materials. We corrected in the manuscript the mistake

Lines 184-186; Out of C16/20 to C30/35 old concrete, how would you produce blends 1,2 and 3 RCA materials? Was the maximum size of RCA 16mm? Authors need to refer any gradation test standards adopted in this test.

The RCA for the blends was crushed with application of the skid-mounted impact crusher on the demolition site. The crushed material was later sieved in the laboratory. Obtained fractions were used to compose the three blends. The sieve analysis was performed in accordance to ISO 17892-4:2016 - Geotechnical investigation and testing - Laboratory testing of soil - Part 4: Determination of particle size distribution.

Authors did not explain in detail what test procedures has been altered.

The alteration of the permeability test was introduced between lines 238-255 in attached manuscript. Each modification was preceded by (1)-(5) sign.  

Authors may check hexavalent chrome limitation required for dam application. Authors explained this in lines 170-171, saying that “Concentration of heavy metals exceeds the limit only in the case chrome.” If this RCA sample does exceed the limit, it must not be used for the dam application.

In this study we did not check the chromium leaching from RCA. Nevertheless, we mentioned this problem in conclusion paragraph. The hexavalent chromium is indeed a great concern and we would like to thank the reviewer for pointing this out. Will focus on this problem in our further leaching tests.

If it does not follow the Darcian rule, I guess we may not use the constant head method.

The constant head method is used for determine the permeability of granular soils with no or little amount of silt fraction. This method is also common in road engineering and is employed especially for soils with disturbed soil skeleton or for reconstituted soil samples. Therefore we decided to use this method in our study. The alternative method – falling head permeability test is planned to be used in  order to characterise the threshold gradient phenomena.

Table 1. Chemical composition of raw materials used in the production of concrete. I am not sure whether this referred data is typical for a normal weight concrete. If this is one of any data, authors may explain this in the text. In addition, RCA needs to be included in Table 1.

The presented data concerns exemplary concrete obtained from mix of cement, fly ash and soil grains. Our idea was to show chemical composition of the concrete, which might be after demolition, used as a source of RCA. The presented data is typical for specified concrete but is not for concrete in general. The appropriate explanation was applied in the manuscript.

Unnecessary concluding remarks need to be removed, at least i.e. 12, 14,…

The unnecessary conclusions was removed and appropriate mention about another chemical compound concentration impact was added.

Lines 37; what does active composting mean?

Active composting is a term used for compost piles which are under storage activities. We changed this term simply to composting.

Lines 56-57; “Designers and contractors are afraid of use recycled aggregates, because their attached and aware of the natural aggregate behavior.” Please rewrite this.

The sentence was rewritten, now is: “Designers and contractors are wary of use recycled aggregates, because the physical and mechanical properties are different from the natural aggregate behaviour.”

Lines 90-91; “Researchers still not fully explained cohesion phenomena which sometimes appears in RCA.” Rewrite this.

The sentence was rewritten, now is:  “The cohesion phenomena which was reported during sear strength tests under RCA is still not fully explained.”

Limes 179; “ … in order to examined potential environmental pollution resulting from use the RCA in mentioned construction, laboratory leaching properties of RCA was determined.” Please rewrite this.

The sentence was rewritten, now is:  “Moreover, laboratory leaching tests of RCA was performed to determine potential environmental pollution.

Thank you for your remarks to this manuscript.

Sincerely,

Andrzej, Wojciech, Justyna, Emil, Alojzy.

Reviewer 2 Report

The topic is of interest and the paper is generally well prepared. Perhaps the part dedicated to the evaluation of the risk of leaching of the heavy metals from RCA is not necessary, as this question is well recognized and it is commonly known that the concrete ensures the excellent immobilization of such substances. I can agree, however, that this part is not large and gives some completion to the main issue that is the problem of permeability. I also find the paper valuable from the practical point of view, as the use of recycled materials in construction becomes still more important today. Finally, I recommend to publish the paper in the journal Applied Sciences.

Author Response

Respond to Review Report 2 for submission of a paper to Applied Sciences MDPI journal

Andrzej Głuchowski, Wojciech Sas, Justyna Dzięcioł, Emil Soból, Alojzy Szymański
Warsaw University of Life Sciences, Faculty of Civil and Environmental Engineering
166 Nowoursynowska str. 02787 Warsaw, Poland

20.11.2018

Dear Reviewer,

We wish to thank You for the remarks. We hope that our further publications will also be appreciated by you.

Sincerely,

Andrzej, Wojciech, Justyna, Emil, Alojzy.

Reviewer 3 Report

The manuscript by Gluchowski et al. entitled “Permeability and leaching properties of recycled concrete aggregate as an emerging material in civil engineering” deals with the experimental characterization of Recycled Concrete Aggregate (RCA) as a substitute for natural aggregate. The manuscript, which focuses on leaching and permeability, provides results from laboratory tests (permeameter). The permeameter device uses an upgraded constant head method. A main conclusion is that the tested RCA blends have adequate permeability properties for being used in civil engineering works (e.g. earth dam construction and roads), and that the concentration of dangerous substances is admissible.

The research target matches the journal aims and scope (applied engineering), and it focuses on a relevant an active research field: the use of eco-friendly materials. The abstract summarizes the main objectives and conclusions. The background is adequate, and the work is both well structured and written. The laboratory tests are adequate and the results are really interesting and useful.However, the following changes/corrections are required (all the questions must be answered within the manuscript):

·      Please, provide the adequate references/cite for the following data: “…the demolition of existing structures generates about 868.5 million tons of waste, which gives 34.7% of total waste production…”, “…at that time, 17.0 million tonnes…”, “around 30,000 tons of aggregate are used to create 1 km of new road”, “…about 88% of the market demand”, “…the demand for recyclable aggregates increased from 5% to 8%...).

·      Line 186: How has been estimated the compressive strength?

·      It is necessary to provide a more comprehensive description of the permeameter device (Section 3.2), in terms of dimensions and materials.

·      Please, use SI units.

 Author Response

Respond to Review Report 3 for submission of a paper to Applied Sciences MDPI journal

Andrzej Głuchowski, Wojciech Sas, Justyna Dzięcioł, Emil Soból, Alojzy Szymański
Warsaw University of Life Sciences, Faculty of Civil and Environmental Engineering
166 Nowoursynowska str. 02787 Warsaw, Poland

20.11.2018

Dear Reviewer,

We wish to thank You for all remarks. We carefully have read your review report and we prepared answers to your comments. The respond to this review ate accompanied by appropriate changes in the manuscript according to Your comments.

Please, provide the adequate references/cite for the following data: “…the demolition of existing structures generates about 868.5 million tons of waste, which gives 34.7% of total waste production…”, “…at that time, 17.0 million tonnes…”, “around 30,000 tons of aggregate are used to create 1 km of new road”, “…about 88% of the market demand”, “…the demand for recyclable aggregates increased from 5% to 8%...).

The appropriate references was introduced in the text. The data was obtained from the Eurostat data and European Aggregate Association Report.

Line 186: How has been estimated the compressive strength?

The data concerning strength class was obtained from the demolished building plans.

It is necessary to provide a more comprehensive description of the permeameter device (Section 3.2), in terms of dimensions and materials.

The additional dimensions of inner and outer cylinders was: inner cylinder dimensions: height h = 0.17 m, diameter d = 0.205 m, outer cylinder dimensions: h = 0.27 m, d = 0.19 m. All parts of the permeameter device are made from stainless steel. The appropriate explanation was provided in the manuscript.

Please, use SI units.

The dimensions of the permeameter and the data about RCA was changed into basic SI units.

Thank you for your remarks to this manuscript.

Sincerely,

Andrzej, Wojciech, Justyna, Emil, Alojzy.

Round  2

Reviewer 1 Report

1) sear strength --> shear strength.

Author Response

Dear reviewer,

Thank You for your comments and remarks.

Sincerely,

Andrzej, Wojciech, Justyna, Emil and Alojzy